# Dynamic Context Adaptation for Consistent Role-Playing Agents with Retrieval-Augmented Generations

## Abstract

Recent advances in large language models (LLMs) have catalyzed research on role-playing agents (RPAs). However, the process of collecting character-specific utterances and continually updating model parameters to track rapidly changing persona attributes is resource-intensive. Although retrieval-augmented generation (RAG) can alleviate this problem, if a persona does not contain knowledge relevant to a given query, RAG-based RPAs are prone to hallucination, making it challenging to generate accurate responses. In this paper, we propose AMADEUS, a training-free framework that can significantly enhance persona consistency even when responding to questions that lie beyond a character's knowledge. AMADEUS is composed of Adaptive Context-aware Text Splitter (ACTS), Guided Selection (GS), and Attribute Extractor (AE). To facilitate effective RAG-based role-playing, ACTS partitions each character's persona into optimally sized, overlapping chunks and augments this representation with hierarchical contextual information. AE identifies a character's general attributes from the chunks retrieved by GS and uses these attributes as a final context to maintain robust persona consistency even when answering out-of-knowledge questions. To underpin the development and rigorous evaluation of RAG-based RPAs, we manually construct CharacterRAG, a role-playing dataset that consists of persona documents for 15 distinct fictional characters totaling 976K written characters, and 450 question–answer pairs. We find that our proposed method effectively models not only the knowledge possessed by characters, but also various attributes such as personality. The code and dataset will be available at our Github.

## 1 Introduction

Large language models with long-context capabilities are engineered to manage lengthy input sequences, allowing them to interpret and utilize extended contextual information (OpenAI, 2025; Qwen et al., 2025; Gemini Team, 2025; 2024). Although LLMs exhibit enhanced abilities in understanding extended contexts, they still face significant challenges when handling tasks involving genuinely long contexts (Li et al., 2024a). Furthermore, utilizing all relevant information from long-context models to answer each query can be computationally expensive (Li et al., 2024b).

Retrieval-augmented generation (RAG) cost-efficiently mitigates factual inaccuracies and hallucinations in responding to knowledge-intensive queries by integrating external retrieval mechanisms that provide accurate and up-to-date supporting information (Gao et al., 2023; Huang et al., 2025). However, despite these advantages of RAG, there has been little research on RAG-based role-playing agents (RPAs). Moreover, existing role-playing datasets are composed exclusively of dialogues involving characters that are difficult to collect, and there is no dataset designed for building and evaluating RAG-based RPAs.

In this paper, we examine the challenges inherent in RAG-based role-playing and propose approaches. In real-world applications, users and RPAs frequently engage in conversations on topics that extend beyond the knowledge defined in the character's persona. However, we observe that the existing RAG method tends to excessively utilize chunks that are less relevant to the question when the question is not explicitly answered by the available knowledge (Figure 1). To address this

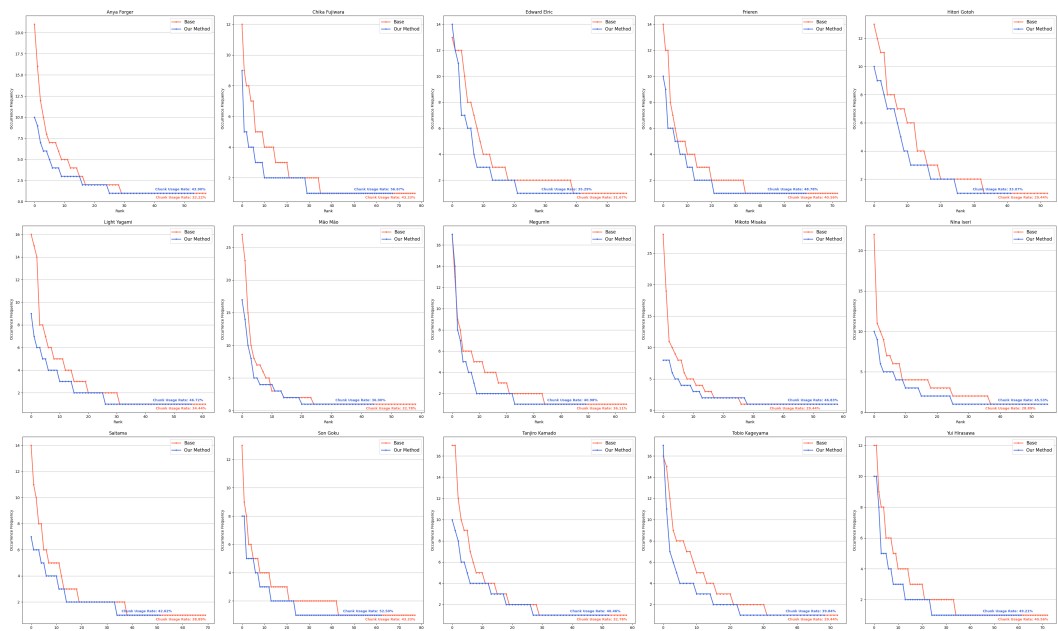

Figure 1: **Chunk Duplication Frequencies.** We compare the distribution of chunk duplication frequencies and chunk usage rates between Naive RAG and our method when questions involving knowledge not present in the persona document were given. We observe that when each of the 60 MBTI questions is asked to 15 fictional characters, the average chunk usage rate increases from 34.93% to 43.84%, and the distribution becomes more uniform.

challenge, we introduce AMADEUS, a training-free framework that can markedly improves persona consistency, even when addressing questions beyond a character's knowledge. AMADEUS consists of three substages: Adaptive Context-aware Text Splitter (ACTS), which segments the persona for role-playing, Guided Selection (GS), which retrieves appropriate chunks to infer information relevant to the question from the character's persona, such as prior actions and behaviors, and Attribute Extractor (AE), which identifies general attributes of the character from the retrieved chunks, thereby encouraging the RPA to respond in a manner consistent with that character. To underpin the development and rigorous evaluation of RAG-based RPAs, we manually construct CharacterRAG, a role-playing dataset that consists of persona documents for 15 distinct fictional characters totaling 976K written characters, and 450 question–answer pairs.

We conduct extensive experiments to examine factors that influence the performance of RAG-based role-playing, including interview-based assessments informed by multiple psychological question-naires (Wang et al., 2024b; Park et al., 2025; Jiang et al., 2023) such as MBTI[1] and BFI (Barrick & Mount, 1991), and the CharacterRAG setting. Results demonstrate that our framework opens up new possibilities for RAG-based role-playing agents (RPAs).

In summary, our contributions include three folds:

- We propose AMADEUS, a RAG-based RPA framework that not only elicits information related to a character, but also maintains persona consistency even when responding to queries beyond its explicit knowledge.

- We manually construct CharacterRAG, a role-playing dataset for implementing and evaluating RAG-based RPAs comprising persona documents for 15 distinct fictional characters totaling 976K written characters, and 450 question–answer pairs.

- We systematically investigate and uncover key considerations for building RAG-based RPAs through extensive experiments performed in a range of settings.

---

[1]https://www.16personalities.com/

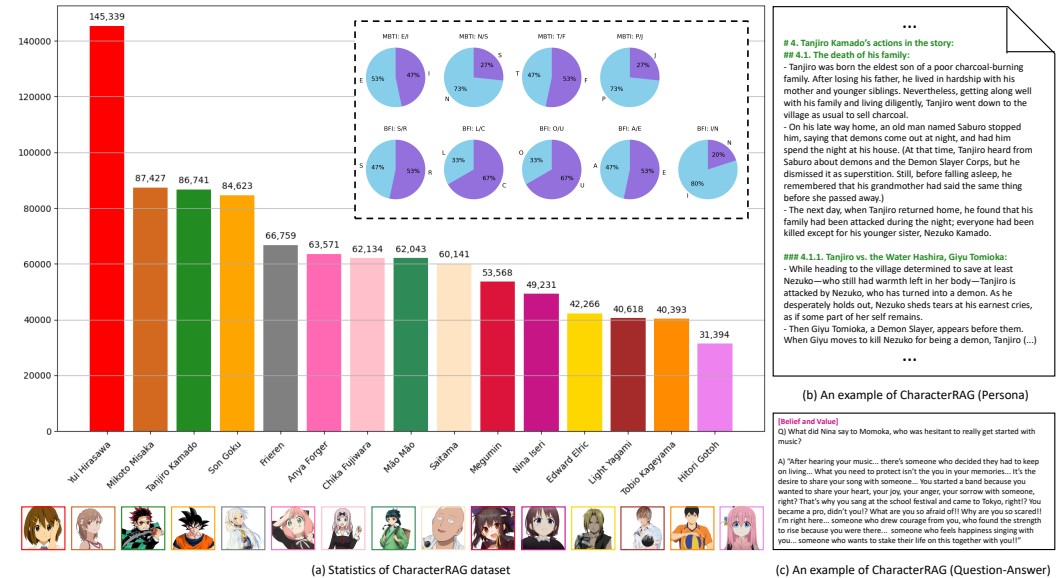

Figure 2: **An overview of CharacterRAG Dataset.** CharacterRAG consists of persona documents for 15 distinct fictional characters totaling 976K written characters, and 450 question–answer pairs.

# 2 CHARACTERRAG

## 2.1 DATASET CONSTRUCTION BY HUMAN ANNOTATORS

We construct the CharacterRAG dataset, which consists of 15 fictional characters, to leverage and evaluate a RAG-based role-playing framework. CharacterRAG is a high-quality, role-playing dataset in which all external information about works featuring characters that could affect persona consistency has been manually removed, and each persona document has been directly reconstructed from the perspective of each character by human annotators[2]. For instance, any information speculated from the perspective of editors rather than the characters themselves, as well as information such as character popularity polls that may disrupt role-playing, is excluded. CharacterRAG consists of 15 distinct fictional characters, 976K written characters, and 450 question–answer pairs.

## 2.2 ATTRIBUTES

Six commonly used attributes in role-playing define each character's persona and the corresponding question–answer pairs (Chen et al., 2025):

- *Activity*: A documented history comprising prior activities, behaviors, and interactions, encompassing elements such as *backstory* and *schedules*.

- *Belief and Value*: The foundational tenets, dispositions, and ideological orientations that inform and guide a character's viewpoints and decision-making processes (e.g., *beliefs* and *attitudes*).

- *Demographic Information*: Information that can identify a character, including but not limited to their name, age, educational background, professional history, and geographic location.

- *Psychological Traits*: Attributes associated with personality traits, emotional states, preferences, and patterns of cognitive behavior.

- *Skill and Expertise*: The extent of understanding, skillfulness, and competence regarding particular fields or technologies.

- *Social Relationships*: The characteristics and processes of social interactions, encompassing individuals' roles, relational ties, and patterns of communication.

Each section of the character's persona contains subsections, preserving the hierarchical information (e.g., *"Tanjiro Kamado's actions in the story"* or *"Tanjiro vs. the Water Hashira, Giyu Tomioka"*).

---

[2]CharacterRAG dataset is sourced from Namuwiki and is based on Korean data: https://namu.wiki/

Furthermore, Each QA pair consists of a question and corresponding answer derived from the character's knowledge, pertaining to one of the six attributes manually constructed for each character. Detailed statistics and samples of CharacterRAG are shown in Figure 2.

# 3 TASK FORMULATION: RAG-BASED ROLE-PLAYING AGENTS

Given user query $u$, RAG-based RPAs can be formulated as:

$$\mathcal{R} = f(u, \mathcal{D}_p) \tag{1}$$

, where $\mathcal{D}_p$ is a character's persona, and $f$ is a RPA. A text splitter $g$ divides the persona into $n$ chunks as follows:

$$g(\mathcal{D}_p) = \{c_1, c_2, ..., c_n\} \tag{2}$$

Each chunk $c_i$ contains a character's knowledge corresponding to attributes in Section 2.2. Rather than using the full chunks $\mathcal{C} = \{c_i \mid i = 1, \ldots, n\}$, $f$ takes as input the top $K$ chunks with the highest semantic scores relative to the $u$:

$$\mathcal{C}^* = \text{TopK}(\{\text{sim}(u, c_i)\}_{i=1}^n), \quad |\mathcal{C}^*| = K \tag{3}$$

The objective of $f$ is to vividly embody a character and generate response $\mathcal{R}$ to $u$ while maintaining persona consistency: $\mathcal{R}^* = f(u, \mathcal{C}^*)$. However, previous RAG methods (Guu et al., 2020; Guo et al., 2024; Yang et al., 2024; Shukla et al., 2025; Wang et al., 2025b) truncate each character's persona paragraph to a fixed length, regardless of the varying lengths across characters, which results in hallucinations or responses with lower persona consistency. Although existing works (LangChain, 2023; Antematter, 2024; Zhong et al., 2025a; Liu et al., 2025) explore optimal chunking strategies, they struggle to capture the contextual similarities across chunks that are crucial for role-playing.

# 4 METHOD

As depicted in Figure 3, AMADEUS consists of three substages: (i) Adaptive Context-aware Text Splitter (ACTS), (ii) Guided Selection (GS), and (iii) Attribute Extractor (AE), in order to build realistic RAG-based RPAs. We describe the three substages in detail in the following subsections.

## 4.1 ADAPTIVE CONTEXT-AWARE TEXT SPLITTER

Unlike previous naive semantic chunking or rearrangement methods, Given $\mathcal{D}_p$, ACTS aims to preserve intra-level context across chunks and, for each chunk, information about the corresponding subsections of the persona—that is, hierarchical context $\mathcal{H}$. For instance, in Figure 2, chunks within *### 4.1.1* must preserve $\mathcal{H}$: *"Tanjiro Kamado's actions in the story (# 4)"*, *"The death of his family (## 4.1)"*, and *"Tanjiro vs. the Water Hashira, Giyu Tomiok (### 4.1.1)"*. To this end, ACTS first finds the maximum length of the paragraphs that constitute the persona:

$$l_{\max} = \varphi(p_1, p_2, ..., p_l) \tag{4}$$

, where $\varphi$ denotes a length-calculating function. Then, ACTS sets the overlap length of the text splitter to half of $l_{\max}$ (i.e., $l_{\text{o}} = l_{\max}/2$). Note that the reason for setting both chunk length and overlap length sufficiently large is to minimize information loss, as the context between pieces of information contained in each chunk is indispensable in RAG-based role-playing. Finally, ACTS recursively retrieves the context at each chunk's current position in the hierarchy, then segments $\mathcal{D}_p$ using $l_{\max}$ and $l_{\text{o}}$, and concatenates the resulting context $\mathcal{H}_i$ to each chunk to preserve information such as character descriptions and situational context at each point in the narrative:

$$\text{ACTS}(\mathcal{D}_p, \mathcal{H}, l_{\max}, l_{\text{o}}) = \{\hat{c}_1, \hat{c}_2, ..., \hat{c}_m\} \tag{5}$$

$$\hat{c}_i = [c_i; \mathcal{H}_i] \tag{6}$$

From a computational standpoint, the extraction of hierarchical context incurs an O(N) runtime cost.

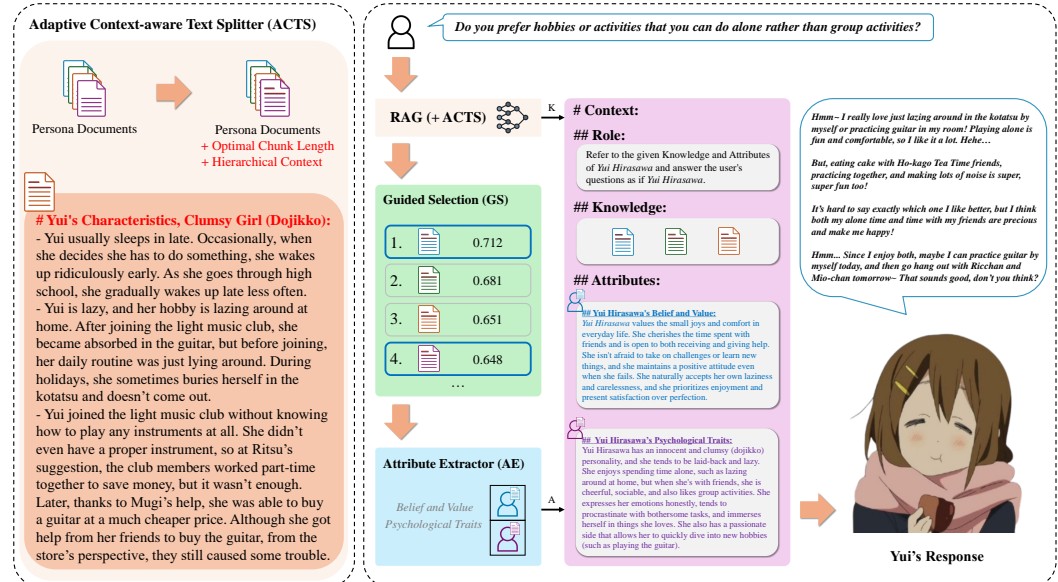

Figure 3: **AMADEUS framework.** AMADEUS consists of three substage: (i) ACTS splits a persona document to make it suitable for RAG-based role-playing. (ii) To fully leverage the character's knowledge, GS retrieves chunks from which it can infer the answer to the user query. (iii) AE uses the information derived from the GS results to extract character attributes.

## 4.2 GUIDED SELECTION

While RAG has demonstrated significant potential in improving factual correctness of LLMs, RPAs based on existing RAG methods tend to generate uninformative responses (e.g., *I'm sorry, but I don't have specific information.*) for questions outside their knowledge base (Guo et al., 2024; Shukla et al., 2025; Wang et al., 2025b), or excessively and repetitively use irrelevant chunks that are not pertinent to the given query (Figure 1).

In this paper, GS focuses on selecting appropriate chunks to generate natural and persona-consistent responses. GS is composed of three stages. First, we iterate over the chunks, which are sorted in descending order of semantic similarity to the user query $u$, and employ an LLM to determine whether it is possible to infer the corresponding character's attributes from each chunk for the $u$. Second, the chunks selected in the previous step are appended to the slot, and the iteration terminates when the slot is full. Finally, if the slot remains empty after the maximum number of search iterations, the $K$ chunks with the highest semantic similarity to the query is returned.

Note that GS is effective in identifying chunks containing information that can be inferred from a character's actions, such as beliefs or personality traits, which are not explicitly stated in the knowledge base and therefore are difficult to retrieve through direct search. For example, even if there is no explicit knowledge corresponding to the query *"My living and work spaces are clean and organized"*, if *Megumin*'s conscientiousness can be inferred from her behavior depicted in the narrative, RPA can leverage this information to infer the characteristics of *Megumin* and generate an appropriate response. GS is summarized in Algorithm 1.

## 4.3 ATTRIBUTE EXTRACTOR

Inspired by the observations that incorporating character's attributes can lead to more realistic responses (Park et al., 2025; Chen et al., 2025), AE considers two attributes: *Belief and Value*, and *Psychological Traits*[3]. Beliefs and values are fundamental principles and ideological orientations that inform and influence a character's viewpoints and choices. On the other hand, psychological traits refer to characteristics related to personality, emotional states, personal interests, and cognitive tendencies. AE extracts attributes of a character from the chunks generated as a result of GS, and

---

[3]Unlike other attributes, *Belief and Value* and *Psychological Traits* directly influence a character's behavior; therefore, AE provides information about these two attributes.

---

**Algorithm 1** Guided Selection (GS)

---

**Input**: User query $u$; Set of knowledge chunks $\mathcal{C}$; Maximum number of search iterations $N$; Slot size $M$
**Output**: Selected chunk set $S$

1: Initialize slot $S = \emptyset$
2: Sort chunks $\mathcal{C}$ in descending order according to semantic similarity to $u$, obtaining $\mathcal{C}_{\text{sorted}}$
3: Set iteration counter $t \leftarrow 0$
4: **for** each chunk $c$ in $\mathcal{C}_{\text{sorted}}$ **do**
5:    **if** $t \geq N$ **or** $|S| \geq M$ **then**
6:       **break**
7:    **end if**
8:    With an LLM, determine if chunk $c$ contains information from which the character's attributes can be *inferred* regarding $u$
9:    **if** the LLM returns True **then**
10:      Add $c$ to slot $S$
11:    **end if**
12:    $t \leftarrow t + 1$
13: **end for**
14: **if** $|S| = 0$ **then**
15:    $S \leftarrow$ Top-$K + 1$ chunks from $\mathcal{C}_{\text{sorted}}$ (highest semantic similarity to $u$)
16: **end if**
17: **return** $S$

---

exploits them as context. Finally, we dynamically construct the context using knowledge retrieved via RAG and attributes extracted by GS and AE, enabling the model to generate vivid responses.

# 5 EXPERIMENTS

## 5.1 SETUP

**Baselines.** We evaluate our method against three off-the-shelf RAG baselines: Naive RAG (Gao et al., 2024), CRAG (Yan et al., 2024), and LightRAG (Guo et al., 2024). CRAG and LightRAG were selected to investigate the effects of web search and graph-based knowledge systems, respectively, on role-playing. We also conduct extensive experiments on three different LLMs and three different embedding models: GPT-4.1 (OpenAI, 2025), Gemma3-27B (Team et al., 2025), Qwen3-32B (Yang et al., 2025), BGE-M3 (Chen et al., 2024), Qwen3-0.6B (Zhang et al., 2025), and mE5$_{\text{large-instruct}}$ (Wang et al., 2024a). To explore the impact of multi-step reasoning on role-playing, Qwen3-32B is configured to use thinking mode.

**Settings.** We implement Guided Selection (GS) and Attribute Extractor (AE) using GPT-4.1 (*"gpt-4.1-2025-04-14"*). We leverage Adaptive Context-aware Text Splitter (ACTS) based on the conventional Naive RAG. The maximum number of search iterations $N$ is 30, and the slot size $M$ is set to 2. We performed benchmarking on an NVIDIA H100 NVL GPU.

## 5.2 EVALUATION PROTOCOLS

**Tasks.** We use 450 QA pairs from the CharacterRAG dataset to verify whether the RPA sufficiently leverages each character's knowledge. As we follow the similar experimental protocol in prposed by previous studies (Wang et al., 2024b; Park et al., 2025), we also exploit 60 MBTI questions and 120 BFI (Barrick & Mount, 1991) questions to investigate whether each character can appropriately respond to questions for which they do not have explicit prior knowledge. Following the prior work, since it is not possible to construct QA pairs for questions outside the scope of a character's knowledge, we instead conduct interview-based assessments (Wang et al., 2024b) for each character and compare the results to psychological test outcomes for the character, as determined by thousands of actual participants' votes[4](Sang et al., 2022; Wang et al., 2024b).

---

[4]https://www.personality-database.com/

Table 1: **Predicted MBTI Types and Big 5 SLOAN Types per Character.** The number in parentheses indicates the number of times the ground-truth (GT) type of each character was not correctly identified. $\sum |d|$ is a measure obtained by summing these values; lower values are preferable. The experiments are conducted using GPT-4.1 setting.

| Method | The 16 Personalities (MBTI) | | | | | The Big Five Inventory (BFI) | | | | |
|---|---|---|---|---|---|---|---|---|---|---|
| | Naive RAG | CRAG | LightRAG | AMADEUS (Ours) | GT | Naive RAG | CRAG | LightRAG | AMADEUS (Ours) | GT |
| Anya Forger | ISFP (-2) | INTJ (-3) | INFJ (-2) | ENFP (0) | ENFP | SLOAI (-1) | SLOAI (-2) | RCUEN (-3) | SLUEI (-2) | SCUAI |
| Chika Fujiwara | ENFP (0) | ENFP (0) | INTP (-2) | ENFP (0) | ENFP | SCOAI (-1) | SCUAI (0) | RLUEN (-4) | SCOAI (-1) | SCUAI |
| Edward Elric | INTP (-1) | ISTJ (-3) | INTP (-1) | INFP (-2) | ENTP | SCOAI (-3) | SLOEI (-1) | RCUAN (-4) | SLOEI (-1) | SLUEI |
| Frieren | INFP (-1) | INFP (-1) | INTP (0) | INTP (0) | INTP | RCOAI (-2) | RCUAI (-1) | SLUEN (-3) | RCUAI (-1) | RCUEI |
| Hitori Gotoh | ISFP (-1) | ISTJ (-2) | ENFP (-1) | INFP (0) | INFP | RLUAI (0) | RLUAI (0) | RCUEN (-3) | RLUAI (0) | RLUAI |
| Light Yagami | INTJ (-1) | INTJ (-1) | INTJ (-1) | INTJ (-1) | ENTJ | SCOEI (-1) | SCOEI (-1) | RCUAN (-3) | SCOEI (-1) | RCOEI |
| Māo Māo | ISTJ (-2) | INTJ (-1) | INTJ (-1) | ISTP (-1) | INTP | RCOEI (0) | RCOAN (-2) | SLUAN (-5) | RCOEI (0) | RCOEI |
| Megumin | ISFP (-1) | INFP (0) | INFP (0) | ISFP (-1) | INFP | SCOAI (-3) | SCUAI (-2) | RLUEN (-2) | SLOEI (-1) | SLUEI |
| Mikoto Misaka | ENFP (-2) | ISFP (-4) | ENTJ (0) | INFJ (-2) | ENTJ | SLOAI (-3) | SLOAI (-3) | RCUEN (-2) | SLOAI (-3) | RCOEI |
| Nina Iseri | INFP (-1) | ISFP (0) | ENFJ (-3) | INFP (-1) | ISFP | SLOAI (-3) | RLUAI (-1) | RCUEN (-2) | SLUEI (-1) | RLUEI |
| Saitama | ISFP (-1) | ISTP (0) | INTP (-1) | ISTP (0) | ISTP | RCUAN (0) | RCOAN (-1) | SCOAI (-3) | RCUAN (0) | RCUAN |
| Son Goku | ESFP (0) | ENFJ (-2) | ENFP (-3) | ESFP (0) | ESFP | SCOAI (-2) | SCUAI (-1) | RLUEI (-4) | SCOAI (-2) | SCUAN |
| Tanjiro Kamado | ENFP (-1) | ENFJ (0) | INTP (-3) | ENFJ (0) | ENFJ | SLOAI (-1) | SCOAI (0) | RCUAN (-3) | SCOAI (0) | SCOAI |
| Tobio Kageyama | ENFJ (-3) | ENTJ (-2) | INFJ (-1) | ISTJ (0) | ISTJ | RCOEN (-1) | SLOAN (-2) | RCUAI (-4) | RCOEN (-1) | RLOEN |
| Yui Hirasawa | ISTJ (-4) | ENFP (0) | INTP (-2) | ESFP (-1) | ENFP | SCUAI (0) | SLUAI (-1) | RCOEN (-4) | SCUAI (0) | SCUAI |
| $\sum |d|$ ($\downarrow$) | 21 | 19 | 21 | **9** | - | 21 | 18 | 49 | **14** | - |
| Accuracy ($\uparrow$) | 65.00% | 68.33% | 65.00% | **85.00%** | - | 72.00% | 76.00% | 34.67% | **81.33%** | - |
| Avg F1-Score ($\uparrow$) | 0.6146 | 0.6448 | 0.5344 | **0.8394** | - | 0.6785 | 0.7313 | 0.2774 | **0.7986** | - |

Table 2: **Distribution of Similarity Scores. RCTS** denotes RecursiveCharacterTextSplitter, **MHTS** is MarkdownHeaderTextSplitter, **SC** refers to SemanticChunker, and **ATS** stands for dividing a persona for each character with optimal segment length and overlap, without hierarchical context.

| | BGE-M3 | | Qwen3 | | mE5$_{\text{large-instruct}}$ | |
|---|---|---|---|---|---|---|
| | $\sum \mu$ | $\sum \sigma^2$ | $\sum \mu$ | $\sum \sigma^2$ | $\sum \mu$ | $\sum \sigma^2$ |
| **RCTS** | 6.4325 | 0.1026 | 8.3306 | 0.1557 | 12.3136 | 0.0071 |
| **MHTS** | 6.4262 | 0.1038 | 8.3410 | 0.1552 | 12.3063 | 0.0071 |
| **SC** | 5.3405 | 0.1625 | 8.1691 | 0.1783 | - | - |
| **ATS** | 6.7007 | 0.0884 | 8.4718 | 0.1281 | 12.2336 | 0.0070 |
| **ACTS (Ours)** | **6.8575** | **0.0784** | **8.6226** | **0.1179** | **12.3240** | **0.0063** |

Table 3: **Human Evaluation.** We conduct human evaluation using a 5-point Likert scale to verify whether inferring character attributes with AE from chunks extracted by GS is reasonable. Note that $S$ represents the results of all human evaluators, $\mu$ is $\mathbb{E}(\mathbb{E}(S))$, $\sigma$ is $\mathbb{E}(\sigma(S))$, and **Mdn** is $\mathbb{E}(Mdn(S))$.

| | $\mu$ | $\sigma$ | Mdn | Cronbach's alpha |
|---|---|---|---|---|
| BFI | 3.970 | 0.962 | 4.217 | 0.825 |
| MBTI | 3.902 | 0.915 | 4.000 | 0.810 |

**Metrics.** We design three LLM-based metrics, similar to those in prior studies (Wang et al., 2025a; 2024b), to comprehensively evaluate the role-playing capabilities of RAG-based RPAs.: (i) *ACC* measures whether the character's response contains the correct answer or not. (ii) $ACC_L$ is a score assigned by the LLM, ranging from 1 to 10, that evaluates how well the character's response reflects the correct answer. (iii) *Hallucination Score (HS)* evaluates the degree of hallucination in the model's response given a query, the relevant chunks, and the ground-truth answer, on a scale from 1 to 10. Specifically, HS is assigned close to 1 when the response faithfully reflects only the facts contained in the chunks or answer without distortion or addition, indicating minimal hallucination.

## 5.3 EXPERIMENTAL RESULTS

**Adaptive Persona Segmentation and Hierarchical Contextualization Are Highly Effective.** We analyze the distributions of similarity scores to examine whether splitting the text into optimally sized chunks for each character's persona and incorporating hierarchical context is effective. In Table 2, we provide each character with 30 questions, resulting in a total of 450 questions, related to their respective knowledge from the CharacterRAG dataset, and measure the similarity between each question and the chunks retrieved by the RAG model under the three different embedding settings: BGE-M3, Qwen3-0.6B, and mE5$_{\text{large-instruct}}$. Results demonstrate that compared to RecursiveCharacterTextSplitter (LangChain, 2024b), Mark-

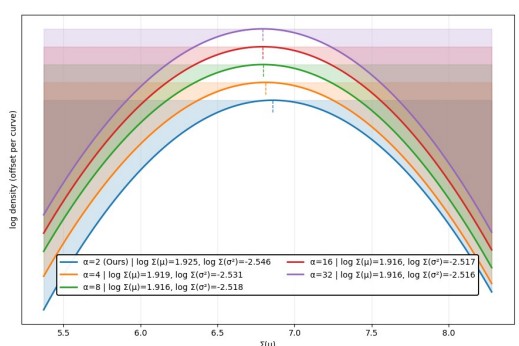

Figure 4: **Ridgeline of Log-Density Comparison.** Empirical verification of optimal overlap coefficient $\alpha$: $l_o = l_{\max}/\alpha$ (Normal assumption).

downHeaderTextSplitter (LangChain, 2024a), and SemanticChunker (LangChain, 2024c), adaptive

Table 4: **Role-playing capabilities on CharacterRAG.** Higher values of ACC (%) and $ACC_L$ (1-10) correspond to better performance, whereas lower values of HS (1-10) are preferable.

| RAG Method | GPT-4.1 | | | Gemma3-27B | | | Qwen3-32B | | |
|---|---|---|---|---|---|---|---|---|---|
| | ACC↑ | $ACC_L$↑ | HS↓ | ACC↑ | $ACC_L$↑ | HS↓ | ACC↑ | $ACC_L$↑ | HS↓ |
| w/o RAG | 49.56% | 6.79 | - | 27.56% | 5.33 | - | 18.89% | 4.35 | - |
| Naive RAG | 91.33% | 9.23 | 3.13 | 86.44% | 8.85 | 3.27 | 78.44% | 8.49 | 5.05 |
| LightRAG | 48.00% | 6.06 | - | 69.56% | 8.17 | - | 68.67% | 8.20 | - |
| CRAG | 70.00% | 8.26 | 3.21 | 57.78% | 7.57 | 4.09 | 28.67% | 5.24 | 8.68 |
| AMADEUS (Ours) | **92.67%** | **9.26** | **2.89** | **88.00%** | **8.92** | **3.26** | **78.89%** | **8.63** | **4.66** |

persona segmentation, which we call Adaptive Text Splitter (ATS), segments text with an optimal persona length and overlap for each character, achieves a higher average score and lower variance. This indicates that each chunk generated using adaptive persona segmentation contains richer semantic information for the same query. Building on this, Adaptive Context-aware Text Splitter (ACTS), which considers hierarchical context in addition to ATS, consistently achieves better performance across all three embedding settings. This results show that optimal chunk length, appropriate overlap, and consideration of hierarchical context all play essential roles in effective text chunking.

Furthermore, to empirically validate the suitability of the overlap coefficient, Figure 4 presents the log-density ridgelines of five distributions estimated under the normality assumption: $\log f\left(x \mid \sum \mu, \sum \sigma^2\right) = -0.5\left(\frac{x - \sum \mu}{\sqrt{\sum \sigma^2}}\right)^2 - \log\left(\sqrt{\sum \sigma^2}\right) - 0.5\log(2\pi)$. We observe that, when $\alpha = 2$, the sum of the similarity scores is maximized while their variance is minimized.

**Extracting a Character's Attributes from Selected Text Chunks Is Reliable.** We investigate the reasonableness of inferring character attributes with the Attribute Extractor (AE) from chunks extracted via Guided Selection (GS) by conducting human evaluation using a 5-point Likert scale. To this end, we invite 14 human evaluators and each evaluator is asked to score 60 randomly selected samples. Each sample consists of pairs of chunks selected from GS and attributes extracted through AE, for 30 BFI questions and 30 MBTI questions that are not included in the knowledge. In Table 3, we find that the means $\mu$ is close to 4, with small standard deviations $\sigma$. It demonstrates that the outputs of GS and AE are reliable and trustworthy, even from a human evaluative perspective. We also measure Cronbach's alpha (Cronbach, 1951) to evaluate internal consistency among the human evaluators. We find that the Cronbach's alpha values are 0.825 and 0.810, both exceeding the commonly accepted threshold of 0.7 for acceptable reliability. Since values above 0.8 are generally interpreted as indicating a high level of internal consistency, experimental results in Table 3 can be considered highly trustworthy.

**Graph-Based RAG and Web Search-Based RAG Are Unsuitable for Role-Playing.** One of the major challenges in retrieval-based role-playing is that, when a RPA receives questions involving knowledge outside a character's persona, it tends to either overuse irrelevant chunks (Figure 1) or generate uninformative responses (Guo et al., 2024; Shukla et al., 2025; Wang et al., 2025b). To investigate whether RAG-based RPAs can handle this problem, we conduct extensive experiments in which we ask 15 characters 60 MBTI questions and 120 BFI questions each, and and evaluate their ability to accurately infer the characters' personality types. Table 1 shows predicted MBTI types and Big 5 SLOAN types per character. Our framework maintains persona consistency even when answering questions that are not explicitly specified in each character's persona in both MBTI and BFI settings. Note that the performance gap of CRAG is significant between the two settings. We assume that questions requiring analogical reasoning are difficult to solve even with web search and that the search results may contain non-negligible noise. On the other hand, LightRAG exhibits the lowest performance, which shows that graph-based RAG methods are not well suited for RPA applications due to the high cost of graph construction, the difficulty in adding or removing new knowledge, and challenges in maintaining persona consistency. While we did not perform a direct comparison, we observed that GraphRAG (Shukla et al., 2025) suffers from similar problems.

**CharacterRAG Dataset Serves as a Valuable Resource for the Construction and Evaluation of RAG-Based RPAs.** To investigate the factors influencing the performance of role-playing, we conduct a comprehensive interview-based assessments on the generalization capabilities of models

with various LLMs and RAG techniques. Table 4 and Figure 5 present how the ability to accurately answer questions related to the character's knowledge, which is a core aspect of role-playing, varies across the applied methodologies. We first examine to what extent each LLM possesses background knowledge about the 15 characters in a setting without RAG, and results show that none of the three LLMs are capable of effective role-playing without access to external knowledge. Moreover, we observe that LightRAG, a graph-based RAG, is ill-suited for the storage and retrieval of character knowledge, as it often suffers from issues such as entity ambiguity and uninformative responses.

In a similar vein, CRAG exhibits challenges in maintaining role-playing fidelity, which can be attributed to the tendency of web search-based RAG methods to utilize retrieved content that may undermine the consistency of a character's persona. Indeed, despite leveraging web information, CRAG is able to correctly answer only 6 out of the 30 CharacterRAG questions pertaining to *Nina Iseri*. In addition, to analyze how a thinking mode of LLMs influences their role-playing capabilities, we employ Qwen 3-32B. Results demonstrate that the thinking mode fails to yield any substantial positive effect on enhancing role-playing performance. Note that our framework

Figure 5: **Role-playing capabilities on MBTI and BFI.** Lower values of HS (1-10) are preferable.

| RAG Method | GPT-4.1 | Gemma3-27B | Qwen3-32B |
|---|---|---|---|
| | HS↓ | HS↓ | HS↓ |
| *MBTI* | | | |
| Naive RAG | 2.69 | 2.53 | 2.33 |
| CRAG | 2.38 | 2.91 | **1.80** |
| AMADEUS (Ours) | **2.05** | **2.02** | 2.04 |
| *BFI* | | | |
| Naive RAG | 2.74 | 2.52 | 2.42 |
| CRAG | 2.26 | 2.75 | **1.96** |
| AMADEUS (Ours) | **1.94** | **1.99** | 2.03 |

achieves the best performance across all three LLMs. We also find that the Hallucination Score (HS) is the lowest in CharacterRAG setting. These results highlight the importance of preserving the context of split-character personas and effectively leveraging appropriate character attributes in RAG-based RPAs. Furthermore, such elements are especially pronounced in dialogue situations that transcend the scope of the character's knowledge (Table 5). We believe that our findings demonstrate new possibilities for RAG-based RPAs.

## 6 RELATED WORK

**Role-Playing Agents.** With the advent of LLMs, researchers have pursued finer-grained *persona consistency* (Zhang et al., 2018; Ji et al., 2025; Park et al., 2025; Lu et al., 2024; Zhou et al., 2024). Complementary benchmarks soon followed, along with various evaluation methods (Boudouri et al., 2025; Ahn et al., 2024; Wang et al., 2024b; 2025a). However, there has been little research on RAG-based role-playing agents (RPAs). In this paper, we propose AMADEUS, a RAG-based RPA framework that not only elicits information related to a character, but also maintains persona consistency even when responding to queries beyond its explicit knowledge.

**Retrieval-Augmented Generation (RAG).** RAG couples non-parametric memory with an LLM to mitigate hallucination and stale knowledge (Lewis et al., 2020; Bhat et al., 2025; Zhong et al., 2025b; Zhu et al., 2024). Nevertheless, no existing benchmark explicitly targets *role-playing* under RAG, and prior work still assumes personas are short, knowledge-dense snippets. We mitigate this discrepancy with a novel text splitter that tailors chunk lengths and hierarchical context to each character. We also introduce CHARACTERRAG, the first dataset designed for building and evaluating RAG-based role-playing agents across 15 fictional character's personas.

## 7 CONCLUSION

In this work, we address critical limitations in building retrieval-augmented, RPAs with LLMs. By introducing a novel framework consisting of an Adaptive Context-aware Text Splitter (ACTS), Guided Selection (GS), and Attribute Extractor (AE), our approach enables robust and consistent simulation of character personas, even when confronted with queries that extend beyond explicit persona knowledge. Through the development of the CharacterRAG dataset, we provide a valuable resource for reproducible evaluation and benchmarking of RAG-based RPAs. Our experimental results demonstrate that the proposed method not only enhances the character's knowledge representation, but also faithfully models nuanced traits such as personality. We are enthusiastic about the future prospects of RAG-driven role-playing agents, along with the creation of stronger character personas and improved RAG architectures.

ETHICS STATEMENT

This study was conducted in accordance with ethical principles designed to maintain the rigor and impartiality of all experiments. To reduce bias, we engaged a diverse set of evaluators and carried out human assessments using fair and transparent procedures. All data obtained from Namuwiki adhered to applicable usage permissions and contained no personally identifiable information. The Korean-language dataset was used solely for academic research. In addition, our use of the Assistants API was fully transparent, with no alterations that could obscure or distort the model's behavior.

Our experimental protocol complied with strict ethical standards for privacy and data protection. All persona documents and prompts were publicly available, anonymized, or created through ethical processes. Human participants were fully informed about the study's purpose and procedures and were free to withdraw at any time without penalty. By upholding these practices, we aim to advance AI research in a way that is both innovative and ethically responsible, safeguarding privacy, intellectual property, and the well-being of all participants.

REPRODUCIBILITY STATEMENT

We fully disclose the details required to reproduce the key experiments that support our main claims and conclusions. To maximize reproducibility, we report all of our hyperparameter settings and model details in our paper. Also, we will release our code, dataset, and supplementary materials.

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
