# OpenReview forum: "Dynamic Context Adaptation for Consistent Role-Playing Agents with Retrieval-Augmented Generations"
_ICLR.cc/2026/Conference — Submitted to ICLR 2026_

### Official Review · Reviewer_gUX3 · 2025-10-22

**Soundness:** 3
**Presentation:** 2
**Contribution:** 2
**Rating:** 4
**Confidence:** 4

**Summary:**

This paper proposes AMADEUS, a training-free RAG framework for role-playing agents that emphasizes persona consistency. It consists of three modules: 1. ACTS (chunking with hierarchical context), 2. GS (LLM-guided gating for personality-rich context), and 3. AE (extraction of beliefs, values, and psychological traits). The authors introduce a new dataset, CharacterRAG, covering 15 anime/game characters with ~976K tokens and 450 QA items, and evaluate consistency using MBTI and BFI personality metrics.

**Strengths:**

- Interesting Setup: Role-playing agents with persona tracking is a novel and popular topic.

- Clear Modular Design: The ACTS → GS → AE pipeline is well-structured and has potential use in other structured attribute extraction tasks.

- New Benchmark: The authors created a sizable benchmark dataset (CharacterRAG), which could contribute to future research in character-centric modeling.

**Weaknesses:**

- Problem Framing – RAG for Personality?
The core task is framed around personality extraction and preservation, which is arguably not well-justified as a role for RAG. Personality is a stable trait, and might be better handled by LLM summarization over full context rather than partial retrieval.

- Evaluation Scope Too Narrow
The evaluation overly focuses on MBTI/BFI-style tests. This does not show if the system performs well in downstream tasks (e.g., in-character QA, stylistic imitation). Also, no evidence is given that this personality focus doesn't degrade performance on other tasks.

- MBTI as Ground Truth?
The paper does not clarify how ground-truth MBTI labels are obtained or validated. Are these canonical? Are there characters with ambiguous or contradictory traits across works?

- Overloaded Character Set
Including 15 characters is impressive but risks overwhelming the reader. It would help to narrow the benchmark or focus deeply on 1–2 characters (e.g., Hirasawa Yui) with full backstory and qualitative insights.

- Lack of Ablation Analysis
No ablations are shown to measure the contributions of ACTS, GS, or AE. In particular, the GS module performs LLM-based reranking—this is a strong boost and may unfairly outperform pure retrieval baselines, confounding the comparisons.

**Questions:**

Why did you choose RAG over LLM summarization for trait extraction, given that personality is a global/stable attribute rather than local/contextual?

How are MBTI ground truths assigned? Are they consensus-based? Are inter-annotator agreements or soft labels considered?

Have you measured whether the AE module introduces hallucinated or speculative traits, especially when context is ambiguous?

What downstream tasks (other than personality QA) can benefit from your pipeline? Could your approach generalize to style transfer or dynamic response generation?

Could you show results using a smaller character subset with detailed breakdowns to improve interpretability?

---

> ### Author Response · Authors · 2025-11-25
>
> # [1/3]
> We would like to express our sincere gratitude to the reviewer gUX3 for their thoughtful comments and valuable suggestions. Below, we provide our responses to each of the points raised.
>
> ```
> Problem Framing – RAG for Personality? The core task is framed around personality extraction and preservation, which is arguably not well-justified as a role for RAG. Personality is a stable trait, and might be better handled by LLM summarization over full context rather than partial retrieval.
> ```
>
> We appreciate the reviewer’s insightful question. To address the reviewer's suggestion, we conduct additional experiments with two baselines: RAPTOR [1] and Adaptive RAG [2]. RAPTOR recursively builds a tree with varying levels of summarization by iteratively applying embedding, clustering (using Gaussian Mixture Model), and summarization steps. Adaptive RAG is reimplemented to include both a module for assessing the relevance of queries and another for detecting hallucinations in responses. It is important to note that we utilize 60 MBTI questions to assess whether RAG-based RPA is capable of accurately responding to queries beyond the limits of its pre-existing knowledge.
>
> |  |  **Naive RAG**  |  **CRAG**  |  **LightRAG**  |  **Adaptive RAG**  |  **RAPTOR**  |  **AMADEUS (Ours)**  |
> | :---: | :---: | :---: | :---: | :---: | :---: | :---: |
> | $\sum \|d\|$ ($\downarrow$) |  21  | 19 | 21 | 15 | 14 | **9** |
> | Accuracy ($\uparrow$) |  65.00\%  | 68.33\% | 65.00\% | 75.00\% | 76.67\% | **85.00\%** |
> | Avg F1-Score ($\uparrow$) |  0.6146  | 0.6448 | 0.5344 | 0.6832 | 0.6925 | **0.8394** |
>
> Results show that methods focused solely on retrieving information, including those that apply LLM summarization, do not achieve the best performance. This is because, as pointed out by the reviewer, previous RAG approaches are not suitable for role-playing where the questions may not be explicitly covered in the retrieved knowledge. These insights have the potential to spark fresh research initiatives within the still emerging area of RAG-based RPA.
>
> ```
> Evaluation Scope Too Narrow The evaluation overly focuses on MBTI/BFI-style tests. This does not show if the system performs well in downstream tasks (e.g., in-character QA, stylistic imitation).  Also, no evidence is given that this personality focus doesn't degrade performance on other tasks.
> ```
>
> As we follow a similar experimental protocol as proposed by previous studies [3, 4, 5], we conducted interview-based assessments for the psychological questionnaires. The advantage of this evaluation method is that we can attain a more nuanced understanding of RPAs’ character fidelity by measuring the personalities of RPAs and comparing them with the personalities of the characters, thereby enabling a systematic and quantitative assessment of how authentically RPAs embody the intended character traits. In future work, we will diversify the downstream tasks used in the benchmark to more comprehensively evaluate the persona consistency of RAG-based RPAs.
>
> ```
> MBTI as Ground Truth? The paper does not clarify how ground-truth MBTI labels are obtained or validated. Are these canonical? Are there characters with ambiguous or contradictory traits across works?
> ```
>
> As shown in Section 5.2, we exploit ground truth as determined by thousands of actual participants’ votes, following existing works [3, 6], because it is not possible to construct QA pairs for questions outside the scope of a character’s knowledge.
>
> ```
> Overloaded Character Set Including 15 characters is impressive but risks overwhelming the reader. It would help to narrow the benchmark or focus deeply on 1–2 characters (e.g., Hirasawa Yui) with full backstory and qualitative insights.
> ```
>
> We appreciate the reviewer’s insightful feedback. In future research, we will not only manually construct personas, but also collect and refine detailed information from the web.
>
> ```
> Why did you choose RAG over LLM summarization for trait extraction, given that personality is a global/stable attribute rather than local/contextual?
> ```
>
> Thanks for this very interesting suggestion! Since even minor information loss in role-playing scenarios (e.g., the omission of full band member names) can significantly impair persona consistency, we opted to employ RAG-based approaches rather than relying on LLM-based summarization.
>
> ```
> How are MBTI ground truths assigned? Are they consensus-based? Are inter-annotator agreements or soft labels considered?
> ```
>
> Similar to previous studies [3, 6], we exploited ground truth labels generated by majority voting among thousands of participants for our experiments.

---

> ### Author Response · Authors · 2025-11-25
>
> # [2/3]
> ```
> Lack of Ablation Analysis No ablations are shown to measure the contributions of ACTS, GS, or AE. In particular, the GS module performs LLM-based reranking—this is a strong boost and may unfairly outperform pure retrieval baselines, confounding the comparisons.
> ```
>
> | Method |  $\sum \|d\|$ ($\downarrow$)  |  Accuracy ($\uparrow$)  |  Avg F1-Score ($\uparrow$)  |
> | :---: | :---: | :---: | :---: |
> | Naive RAG |  21  |  65.00\%  |  0.6146  |
> | Naive RAG + ACTS |  19  |  68.33\%  |  0.6524  |
> | Naive RAG + ACTS + GS |  15  |  75.00\%  |  0.7426  |
> | **Naive RAG + ACTS + GS + AE (Ours)** | **9**  |  **85.00\%**  |  **0.8394**  |
>
> The table above demonstrates the results of ablation studies using GPT-4.1 ("gpt-4.1- 2025-04-14"), evaluated on 60 MBTI questions. We observe that ACTS, GS, and AE all perform well when responding to queries outside a character’s knowledge scope. AE leads to the greatest improvement in performance, suggesting that incorporating extracted character attributes as inputs is substantially more beneficial than just adding chunks linked to the user’s query.
>
> Though LLM-based reranking methods are strong baselines [1, 2], simply relying on such approaches does not lead to significant performance improvements.
>
>
> ```
> Have you measured whether the AE module introduces hallucinated or speculative traits, especially when context is ambiguous?
> ```
>
> To verify the validity of the AE in extracting attributes from the chunks selected by the GS, we invited 14 human evaluators, and each evaluator was asked to score 60 randomly selected samples. Table 3 shows that the means $\mu$ is close to 4, with small standard deviations $\sigma$. We further evaluated the consistency of our human evaluators using Cronbach’s alpha, obtaining scores of 0.825 and 0.810. Both values are above the standard cutoff of 0.7, demonstrating acceptable reliability.
>
> ```
> What downstream tasks (other than personality QA) can benefit from your pipeline? Could your approach generalize to style transfer or dynamic response generation?
> ```
>
> One of the major challenges lies in approximating the linguistic style of the character. This is a critical issue that must be addressed in RAG-based role-playing frameworks, and we will focus on this problem in our future research.
>
> ```
> Could you show results using a smaller character subset with detailed breakdowns to improve interpretability?
> ```
>
> Sure! Here's an example.
>
> ### **[Question]**
> - “Are you confident that everything will work out?”
>
> ### **[Knowledge Chunks]**
> \# Yui's Characteristics and Personality:
>
> - Yui appears to be quite popular with her classmates, probably because she’s kind and makes friends easily. However, she can also be excessively careless and oblivious to the point that she sometimes causes trouble for others without even realizing it or worrying about it.
> - Despite her overwhelming airheadedness, Yui still gets along well with her friends. At her core, she is good-natured, cheerful, and kind to everyone.
> - Nodoka once said, “Yui’s caused more trouble than I can describe, but seeing her innocent smile just puts me at ease,” and the other Light Music Club members agreed with her.
> - Yui has a laid-back, go-at-your-own-pace personality and is not easily embarrassed—even when wearing clothes that might make others uncomfortable. Thanks to this, she willingly goes along with Sawako Yamanaka’s cosplay hobby.
>
> \# Yui's Relationships: Akiyama Mio:
>
> - Mio consistently ranks within the top 10 of her entire grade, excelling in all subjects.
> - She is shy and dislikes being the center of attention, easily gets lonely, has a weak stomach, and is often timid. Mio is particularly sensitive and gets extremely scared by creepy stories or anything to do with ghosts.
> - Mio became good friends with Hirasawa Yui as well. Yui acts as a sort of catalyst for Ritsu, teaming up with her to play pranks on Mio—making the pranks even more powerful. Unlike with Ritsu, though, Mio never lays a hand on Yui, even when they’re goofing around together. She'll only hit Ritsu, not Yui. Although it’s rare for Yui to prank Mio, when she does, Mio is completely defenseless against it, much more so than with Ritsu.
>
> \# Yui’s Whereabouts, Season 2, Extra Episode 1 "Planning Meeting!":
>
> - After showing Ton’s fish tank, Ritsu boasts that since joining the light music club, she has gotten accepted by her preferred university, grown taller, and her skin has become fair and clear. Ritsu’s face is not fully shown; her eyes are obscured. Next, a glimpse is given of a plastic frog dressed in an outfit made by Sawako.
> - Two first-year students who are fans of Mio are interviewed and praise Mio, saying the live performance was amazing.
> - Yui’s next-door neighbor, Grandma Ichimonji Tomi (一文字とみ), also says she’s looking forward to the talent show next year.

---

> ### Author Response · Authors · 2025-11-25
>
> # [3/3]
> ### **[GS Output]**
> \# Yui’s Whereabouts, Season 2, Extra Episode 3 “Keikaku!” - Plan! Episode 27. Plan!:
> - During this time, Azusa was worried about whether it was okay that everyone had the college entrance exams coming up soon, but Yui and Ritsu confidently assured her that, of course, they would do fine on the exams. Yui said that she could feel motivated as long as there were snacks, and Mio agreed with a similar opinion, leading Azusa to mutter to herself that they were just escaping from reality. In response, Yui and Ritsu insisted that they were actually working hard, and Mio also mentioned that taking it easy every now and then wasn’t a bad thing. Azusa looked at Mio and thought to herself that Mio must actually be pretty worried, too.
> - After buying a guidebook, Azusa said she wanted to go to New York (ニューヨーク), and Yui made a pun, asking if Azusa meant she wanted to take a bath (入浴, which is pronounced the same way), making an old-man joke before imagining that Gita’s hometown must be New York as well.
>
> \# Yui’s Actions, Season 2, Episode 12 “Natsu Fesu!” – Summer Festival!
> - When Yui said that their own performance was even more amazing, Ritsu was surprised and asked how she could be so confident when comparing themselves to professional musicians, telling her to at least practice playing guitar without making mistakes before saying things like that. But Yui replied, “But I think we’re amazing because we’re together.” Mio and Azusa agreed with Yui, and Mio added that next time, they should come back not just to watch, but to perform. Ritsu said it would be truly incredible if that could happen, and everyone agreed with Yui’s wish that they could always be together.
>
> ### **[AE Output]**
> \#\# Hirasawa Yui's Belief and Values:
> - Hirasawa Yui has a positive and optimistic outlook, firmly believing that "anything can be achieved if we do it together." Even when facing difficult tasks or important exams, she approaches them with confidence and the positive expectation that "of course I can do well" and "things will work out if I try my best." She treasures the time spent with her friends, valuing community and friendship. Rather than worrying about reality, she prioritizes the enjoyment of the present, teamwork, and the importance of effort.
>
> \#\# Yui Hirasawa's Psychological Traits:
> - Yui Hirasawa is innocent, cheerful, optimistic, and highly imaginative. She is honest in expressing her emotions, has a good sense of humor, and possesses a simple-mindedness that allows her to find joy in even the smallest things. Yui tends to focus more on the fun and positive aspects of life rather than worries or anxieties. She is affectionate in her relationships with friends and values her sense of belonging within a group. Even in important moments like exams or performances, she shows a positive and easygoing mindset, often believing and saying, "It’ll be all right," instead of feeling nervous.
>
> ### **[Response (Yui Hirasawa)]**
> - **Yui**: "Ehehe\~! For some reason, I always tend to think, “Everything will work out!” Whether it’s a test or a performance, I might get a little worried at first, but in the end, I try to stay positive and think, “It’ll work out somehow!”
> I feel even more reassured when I’m with my friends, like as long as we’re together, we can do anything!
> That’s why I’m always bursting with confidence~☆
> It’s okay to make mistakes! It’s still fun, and I can always do better next time!
> Believing that everything will turn out just fine as long as we all do our best together—that’s the Yui style!"
>
>
> We sincerely appreciate once again for the reviewer's valuable feedback. We hope that our responses have satisfactorily addressed your concerns. Should any additional questions arise or further clarification be needed, further discussion would be welcome.
>
> [1] Sarthi et al., “RAPTOR: Recursive Abstractive Processing for Tree-Organized Retrieval.“ ICLR. 2024
>
> [2] Jeong et al., “Adaptive-RAG: Learning to Adapt Retrieval-Augmented Large Language Models through Question Complexity.“ NAACL. 2024.
>
> [3] Wang et al., “INCHARACTER: Evaluating Personality Fidelity in Role-Playing Agents through Psychological Interviews.” ACL. 2024.
>
> [4] Jiang et al., “PersonaLLM: Investigating the Ability of Large Language Models to Express Personality Traits.” NAACL. 2024. Findings.
>
> [5] Park et al., “CharacterGPT: A Persona Reconstruction Framework for Role-Playing Agents.“ NAACL. 2025.
>
> [6] Sang et al., “MBTI Personality Prediction for Fictional Characters Using Movie Scripts.“ EMNLP. 2022.

---

### Official Review · Reviewer_EE4o · 2025-10-30

**Soundness:** 2
**Presentation:** 2
**Contribution:** 2
**Rating:** 2
**Confidence:** 4

**Summary:**

This paper investigates RAG-based role-playing agents, which aim to answer user queries using information retrieved from persona documents. The authors propose a new retrieval framework specifically designed for hierarchically structured persona documents, consisting of three key components: Adaptive Context-aware Text Splitter (ACTS), Guided Selection (GS), and an Attribute Extractor.

The ACTS module segments text into overlapping chunks while prepending contextual information from parent nodes within the hierarchical structure to each chunk. The GS component filters these chunks by prompting an LLM to verify whether the correct persona information can be inferred from each chunk. Attribute Extractor extracts psychological traits and value/belief-related attributes from the selected text segments.

The authors construct a self-collected dataset named CharacterRAG, containing 15 distinct characters and 450 question–answer pairs. Their experiments are conducted on this dataset.

**Strengths:**

i. This paper's attempt to improve the retrieval accuracy of persona information for RAG-based role-playing agents is meaningful.

ii. The collection of a new dataset demonstrates the authors’ effort to empirically explore this problem and provides a potential resource for future studies.

**Weaknesses:**

i. The paper is poorly written, and many essential details are missing, which makes it difficult to fully understand and reproduce the proposed approach.

- The description of the CharacterRAG dataset construction process lacks sufficient detail. It is unclear how the persona documents were collected and processed, how the 450 QA pairs were generated, and what standards were used to filter unqualified documents. Furthermore, the authors do not discuss any measures taken to ensure the fidelity and correctness of the dataset. Since all experiments rely entirely on this dataset, these details are critical to ensure the fairness and reliability of the results.

- The method descriptions are confusing and incomplete. The paper claims that the Adaptive Context-aware Text Splitter (ACTS) divides text into optimally sized and overlapping chunks; however, Section 4.1 provides no information on how the optimal chunk length is determined or how the overlap is implemented. Similarly, in Section 4.2 (Guided Selection), the details of the LLM prompting process are missing — including how the prompt templates are constructed and how the model’s responses are evaluated or extracted. In Section 4.3 (Attribute Extractor), the authors only explain why and what attributes are extracted, but fail to specify how the extraction is actually performed.

- According to Section 5.2, Table 4 appears to present the main experimental results. However, both the table and its corresponding analysis are placed later in Section 5.3, which disrupts the logical flow of the paper and makes the structure somewhat confusing.

- In Figure 1, the authors do not explain how the chunk duplication frequencies and chunk usage rates are computed, which makes the figure difficult to interpret. In addition, the text in Figures 1 and 2 is too small to read clearly.

- Figure 5 is mislabeled and should actually be Table 5.

ii. The proposed method appears to be highly tailored to the CharacterRAG dataset. The ACTS component relies on the dataset’s hierarchical structure, while the Attribute Extractor is built upon its predefined attribute taxonomy. As a result, the method may lack scalability and generalizability to other datasets or real-world scenarios. Moreover, comparing this approach with other methods on CharacterRAG may not provide a fair evaluation of its effectiveness.

iii. Guided Selection (GS) relies on LLM to filter chunks, computationally expensive and impractical for large-scale persona documents.

iv. In the main experiments presented in Table 4, despite the dataset-specific design efforts, the improvement over the baseline naive RAG model is marginal, which makes the overall contribution appear limited. Although the proposed method performs relatively well on tasks related to MBTI and BFI, the Guided Selection (GS) component explicitly extracts MBTI- and BFI-related attributes (with the implementation details remaining unclear). Therefore, it is unsurprising that the method outperforms the naive RAG under these conditions.

**Questions:**

see the weakness.

---

> ### Author Response · Authors · 2025-11-26
>
> # [1/4]
> We sincerely appreciate the reviewer EE4o's insightful comments and constructive suggestions. Below, we provide detailed responses to each of the points raised.
>
> ```
> The description of the CharacterRAG dataset construction process lacks sufficient detail. It is unclear how the persona documents were collected and processed, how the 450 QA pairs were generated, and what standards were used to filter unqualified documents.  Furthermore, the authors do not discuss any measures taken to ensure the fidelity and correctness of the dataset. Since all experiments rely entirely on this dataset, these details are critical to ensure the fairness and reliability of the results.
> ```
>
> Thanks for raising this question. As shown in Section 2.1, CharacterRAG is sourced from wiki pages. This wiki site allows users to freely contribute content, but they must strictly adhere to the editing guidelines and other rules (e.g., restrictions on prohibited content). We first instructed human annotators to extract each character's wiki page in markdown format, and then asked them to remove any explicitly extradiegetic information that could potentially affect role-playing (e.g., results from character popularity polls, descriptions comparing the character to protagonists from other works, or statements regarding the character’s voice actor). Finally, the authors manually inspected the results to verify that the dataset was constructed in accordance with the established guidelines.
>
> For the 450 QA pairs, human annotators were asked to review the constructed persona for each character and, similar to the persona construction process, generate five QA pairs for each attribute (in Section 2.2), resulting in a total of 30 QA pairs per persona. Subsequently, the authors reviewed which parts included questions and answers. Finally, we included a total of 450 QA pairs, each accompanied by the corresponding character's persona, in the dataset to enable effective evaluation of the RAG-based RPA system’s overall construction and performance. We note that there has been little research on role-playing based on RAG, and therefore we built CharacterRAG with high quality and plan to release it to the public. We believe that CharacterRAG dataset will be greatly beneficial for future works.
>
> ```
> The paper claims that the Adaptive Context-aware Text Splitter (ACTS) divides text into optimally sized and overlapping chunks; however, Section 4.1 provides no information on how the optimal chunk length is determined or how the overlap is implemented.
> ```
> As shown in 4.1, ACTS finds the maximum length of the paragraphs that constitute the persona. At each iteration, ACTS loads each paragraph that constitutes a persona, measure its length, and define the longest length among them as $L_{max}$. Then, in Figure 4, we empirically set overlap $\alpha$. Result demonstrates that each chunk contains the most abundant semantic information when $\alpha$ is set to 2. Note that a large overlap is used to minimize information loss between consecutive chunks, thereby improving persona consistency.

---

> ### Author Response · Authors · 2025-11-26
>
> # [2/4]
> ```
> Similarly, in Section 4.2 (Guided Selection), the details of the LLM prompting process are missing — including how the prompt templates are constructed and how the model’s responses are evaluated or extracted.  In Section 4.3 (Attribute Extractor), the authors only explain why and what attributes are extracted, but fail to specify how the extraction is actually performed.
> ```
>
> Thank you for highlighting this point. We used the prompt from GS as follows:
>
> ```
> [GS Prompt]
> In the given [persona_name]'s Context, output True if you can clearly infer the Belief and Value information or Psychological Traits of [persona_name] to answer the given User Query; otherwise, output False. Output only True or False.
>
> # Explanation of Belief and Value:
> - The principles, attitudes, and ideological stances that shape an individual’s perspectives and decisions
> - e.g., [persona_name]'s "beliefs", "attitudes", "values", "religion"
>
> # Explanation of Psychological Traits:
> - Characteristics related to personality, emotions, interests, and cognitive tendencies.
> - e.g., [persona_name]'s "Personality", "hobby and interest", "emotional"
> ```
>
> We utilize 60 MBTI questions to assess whether the RAG-based RPA is capable of accurately responding to queries beyond its current knowledge base. Table below shows that ACTS, GS, and AE are all highly effective at handling questions that go beyond a character's knowledge. AE demonstrates the most significant enhancement in performance, emphasizing that integrating character attributes as input is considerably more effective than merely including segments associated with the user query.
>
> | Method |  $\sum \|d\|$ ($\downarrow$)  |  Accuracy ($\uparrow$)  |  Avg F1-Score ($\uparrow$)  |
> | :---: | :---: | :---: | :---: |
> | Naive RAG |  21  |  65.00\%  |  0.6146  |
> | Naive RAG + ACTS |  19  |  68.33\%  |  0.6524  |
> | Naive RAG + ACTS + GS |  15  |  75.00\%  |  0.7426  |
> | **Naive RAG + ACTS + GS + AE (Ours)** |  **9**  |  **85.00\%**  |  **0.8394**  |
>
> ```
> In Figure 1, the authors do not explain how the chunk duplication frequencies and chunk usage rates are computed, which makes the figure difficult to interpret. In addition, the text in Figures 1 and 2 is too small to read clearly.
> ```
>
> In Figure 1, previous RAG method retrieves (k = 3) relevant chunks for each of the 60 given MBTI questions, and subsequently measures both the chunk utilization rate and the duplication frequency. Our method measures the chunk utilization rate and duplication frequency for the chunks selected by ACTS and GS. Here’s an example.
>
>
> ### **[Question]**
> Do you feel uncomfortable approaching someone you’re interested in and starting a conversation?
>
> ### **[Naive RAG]**
> - Frequency distribution in Figure 1: [17, 17, 12, 10, ...]
> - **[Used Chunk]**
>
> ```
> <Chunk #1>
> - When his favorite food, angelica tree shoots (tara-no-me), is placed in front of him, he pretends to be indifferent and tries to keep a calm demeanor. (It seems that because he sees himself as the eldest son, he doesn't want to show how excited he is about food.) However, his nostrils flare, so everyone around him can already tell how he really feels.
> - Before joining the Demon Slayer Corps, when Tanjiro would go down to the town to sell charcoal, the townspeople would warmly greet him, calling him "Tanjiro-chan" in a familiar and affectionate way.
>
> <Chunk #2>
> - However, since she (Daki) is now a demon, she says it doesn’t matter what she does anymore. Then, as if thinking that continuing the conversation is meaningless, he says that's enough and overwhelms Daki with his strength.
>
> <Chunk #3>
> - …I'm sorry. You got hurt, didn't you? Are you oka—huh…?
> - If you like Nezuko, you need to act properly. If you keep acting spoiled, she’ll only see you as a "younger brother" and won’t see you as a romantic interest. Nezuko is a kind girl, so if you act needy, she’ll take care of you, but Zenitsu, are you really okay with just being her little brother? Right now, you don’t even have many chances to show off as a swordsman, so you should fix your daily habits first. Besides, if you don’t clearly tell her how you feel, Nezuko probably won’t notice. After hearing this, Zenitsu apparently started waking up early and completely stopped complaining from the next day. Nezuko was actually worried, wondering if something was wrong with him.
> ```
>
> - **[Response (Tanjiro Kamado)]**
>
> **Tanjiro**: "If I’m interested in someone, I truly want to connect with them, so I try to approach them first. Especially if there’s something I can do to help!
>
> So, approaching someone I'm interested in and starting a conversation just comes naturally to me. Of course, if I'm trying to hide my feelings, like when I'm in front of a durup (angelica-tree shoots), it can get a little awkward... I guess my feelings always show—like when my nostrils flare up! Haha."

---

> ### Author Response · Authors · 2025-11-26
>
> # [3/4]
> ### **[AMADEUS]**
> - Frequency distribution in Figure 1: [10, 9, 8, 6, ...]
> - **[Used Chunk]**
>
> ```
> <Chunk #1>
> # Other:
> - Even before his relationships get close, he has a habit of calling people by their names very quickly. In Tanjiro's case, he calls others by name much faster than is typical for Japanese people.
> - Tanjiro is tone-deaf.
> - Although Tanjiro himself is not just articulate but actually a very skilled speaker, he is really bad at teaching or explaining things. For example, when he first trained in Total Concentration: Constant, he tried to explain what he saw, heard, and felt to Zenitsu and Inosuke, but his explanations were so messy and confusing that they were of no help to their learning. And right after his mark appeared, when Kagaya Ubuyashiki asked him to explain it, he ended up repeating odd sound effects and vague descriptions, just like when he was trying to teach Total Concentration: Constant, which left others at a loss. This trait seems to be something he shares with Mitsuri Kanroji.
>
> <Chunk #2>
> # Quotes:
> - …I'm sorry. You got hurt, didn't you? Are you oka—huh…?
> - If you like Nezuko, you need to act properly. If you keep acting spoiled, she’ll only see you as a "younger brother" and won’t see you as a romantic interest. Nezuko is a kind girl, so if you act needy, she’ll take care of you, but Zenitsu, are you really okay with just being her little brother? Right now, you don’t even have many chances to show off as a swordsman, so you should fix your daily habits first. Besides, if you don’t clearly tell her how you feel, Nezuko probably won’t notice. After hearing this, Zenitsu apparently started waking up early and completely stopped complaining from the next day. Nezuko was actually worried, wondering if something was wrong with him.
> - Even if I disappear, I hope these peaceful days will continue. Even if I am no longer by your side...
> ```
>
> - **[Response (Tanjiro Kamado)]**
>
> **Tanjiro**: "Yeah, that’s right. For me, it actually feels more natural to approach someone first when I meet them, call their name, and start a conversation. I think it’s easy to become friends, even with someone I’m meeting for the first time, as long as their heart isn’t closed off. If you treat anyone warmly, they’ll gradually open up to you as well. And if I’m curious about something, I prefer to ask honestly. Rather than feeling awkward or nervous, I just genuinely want to know what kind of person they are, and I hope I can be helpful to them.
>
> If you ever feel like it’s difficult to approach someone at first, don’t worry too much. What’s important is that your sincerity comes through. Even if I’m not always good with words, I try my best to be honest about how I feel. When you want to get close to someone you care about, I believe that showing your true feelings will definitely lead to a good connection!"
>
> - We observe that the existing RAG method tends to excessively utilize chunks that are less relevant to the question when the question is not explicitly answered by the available knowledge. In the camera-ready version, we will add the graph of one character along with the above example for explanation, and include the remaining graphs with increased image size in the appendix to improve interpretability.
>
> ```
> The proposed method appears to be highly tailored to the CharacterRAG dataset. The ACTS component relies on the dataset’s hierarchical structure, while the Attribute Extractor is built upon its predefined attribute taxonomy. As a result, the method may lack scalability and generalizability to other datasets or real-world scenarios. Moreover, comparing this approach with other methods on CharacterRAG may not provide a fair evaluation of its effectiveness.
> ```
>
> Note that there has been little research on RAG-based RPAs, and consequently, there is a lack of datasets available for building and evaluating such RPAs. In this work, we built CharacterRAG with high quality and plan to release it to the public.
>
> The CharacterRAG dataset was fundamentally constructed in markdown format to preserve the hierarchical context necessary for role-playing, and therefore, we do not agree with the feedback that it is not suitable for comparison with other RAG methods. Furthermore, with regard to the comment suggesting that certain text data formats may advantage baseline RAG methods, we have not found convincing supporting evidence in the literature, and therefore cannot fully agree with this perspective.

---

> ### Author Response · Authors · 2025-11-26
>
> # [4/4]
> ```
> In the main experiments presented in Table 4, despite the dataset-specific design efforts, the improvement over the baseline naive RAG model is marginal, which makes the overall contribution appear limited. Although the proposed method performs relatively well on tasks related to MBTI and BFI, the Guided Selection (GS) component explicitly extracts MBTI- and BFI-related attributes (with the implementation details remaining unclear). Therefore, it is unsurprising that the method outperforms the naive RAG under these conditions.
> ```
>
> We disagree that GS focuses on attributes related to MBTI or BFI; rather, its purpose is to identify chunks that can infer answers to a given question when the persona does not possess direct knowledge relevant to the question. Table 4 presents the results of an experiment measuring how well each RAG method answers questions based on the most fundamental knowledge in RAG-based role-playing.
>
> ```
> Guided Selection (GS) relies on LLM to filter chunks, computationally expensive and impractical for large-scale persona documents.
> ```
>
> We assess each approach by recording the total latency, the number of tokens utilized, and the aggregate cost over all 60 MBTI questions. Consistent with the experimental setup documented in the paper, we use GPT-4.1 ("gpt-4.1-2025-04-14") as the underlying model. For RAPTOR [1], the evaluation also accounts for the additional cost associated with constructing the tree structure.
>
> Results demonstrates that although GS effectively utilizes the character’s persona there is still room for further improvement. GS has a failure probability of 15.33% across all search iterations. In terms of GS processing speed, Gemma3-27B requires approximately 0.1 seconds for a single request in the vLLM environment, whereas GPT-4.1 takes around 0.49 seconds. The overall latency measured for LightRAG was 25,065 seconds, while to approximately 81,900 seconds for GraphRAG.
>
> |  |  **Naive RAG**  |  **CRAG**  |  **Adaptive RAG [2]**  |  **RAPTOR [1]**  |  **AMADEUS (Ours)**  |
> | :---: | :---: | :---: | :---: | :---: | :---: |
> | Total Latency (s) |  4048.61  | 9,214.99 | 36929.92 | 17155.33 | 17173.02 |
> | Total Tokens |  486058  | - | 12,187,670 | 2,485,627 | 5,649,665 |
> | Total Cost ($) |  1.87  | 6.67 | 28.26 | 9.88 | 14.05 |
>
>
> ```
> According to Section 5.2, Table 4 appears to present the main experimental results. However, both the table and its corresponding analysis are placed later in Section 5.3, which disrupts the logical flow of the paper and makes the structure somewhat confusing.
>
> Figure 5 is mislabeled and should actually be Table 5.
> ```
>
> We sincerely appreciate the reviewer’s valuable and insightful feedback. In the camera-ready version, We will revise the points the reviewer mentioned.
>
>
> Thank you once again for the reviewer’s thoughtful comments. We greatly appreciate the reviewer’s valuable feedback on our work. If the reviewer has any further questions or comments, please feel free to let us know. We would be pleased to discuss them further.
>
>
> [1] Sarthi et al., “RAPTOR: Recursive Abstractive Processing for Tree-Organized Retrieval.“ ICLR. 2024
>
> [2] Jeong et al., “Adaptive-RAG: Learning to Adapt Retrieval-Augmented Large Language Models through Question Complexity.“ NAACL. 2024.

---

### Official Review · Reviewer_DAgg · 2025-10-30

**Soundness:** 3
**Presentation:** 3
**Contribution:** 3
**Rating:** 6
**Confidence:** 4

**Summary:**

This paper tackles hallucination and weak persona consistency in RAG-based role-playing agents (RPAs) when handling queries outside a character’s knowledge, proposing AMADEUS, a training-free framework. AMADEUS includes three key modules: Adaptive Context-aware Text Splitter (ACTS) for hierarchical context-enhanced persona chunking, Guided Selection (GS) for attribute-relevant chunk retrieval, and Attribute Extractor (AE) for key character attribute extraction to sustain consistency. The authors also build CharacterRAG, a dataset with 15 fictional characters’ persona documents (976K characters) and 450 QA pairs, filling the gap of RAG-based RPA-specific evaluation resources. Experiments against baselines (Naive RAG, CRAG, LightRAG) on multiple LLMs demonstrate AMADEUS outperforms others in both in-knowledge and out-of-knowledge scenarios, lowering hallucination and boosting consistency.

**Strengths:**

1. AMADEUS (with ACTS, GS, AE modules) fixes RAG-based RPAs’ hallucinations and poor persona consistency in out-of-knowledge queries, outperforming traditional RAG by enhancing chunking, filtering, and attribute extraction.

2. The manually built CharacterRAG (15 characters, 976K chars, 450 QAs) removes interference (e.g., editor’s inferences) and fills the lack of dedicated RAG-based RPA evaluation resources.

3.  Using 3 LLMs, 3 embedding models, 3 baselines, and covering in/out-of-knowledge scenarios, experiments combine quantitative metrics (ACC, HS) and human evaluation (Cronbach’s α > 0.8) for credible results.

**Weaknesses:**

1. The CharacterRAG dataset includes 15 fictional characters, but the paper does not specify their genre (e.g., anime, novel, film) or personality span (e.g., introverted vs. extroverted, heroic vs. villainous). If characters are concentrated in a single genre or share similar traits, the framework’s generalization to diverse role-playing scenarios (e.g., classical novel characters) remains unvalidated.

2. The Attribute Extractor (AE) only extracts "Belief and Value" and "Psychological Traits," claiming they "directly influence behavior". But it does not explain why other attributes (e.g., "Social Relationships" or "Skill and Expertise," which also shape role responses) are excluded, nor provide comparative experiments to prove these two attributes are more critical for persona consistency. Is it specific to tasks like MBTI and BFI?

3. The evaluation dataset is incomplete. The paper only evaluates tasks like MBTI and BFI, but role-playing involves many other dimensions. Is the method applicable to other role-playing tasks? For example, imitating a character's linguistic style or simulating a character's behavior.

**Questions:**

1. How is L_max set?
2. Why can finding the information of the k most similar chunks solve problems that are beyond the scope of the character's knowledge?

---

> ### Author Response · Authors · 2025-11-25
>
> # [1/2]
> We are grateful for the reviewer DAgg’s thoughtful feedback and valuable suggestions. We address each of the raised points below.
>
> ```
> The CharacterRAG dataset includes 15 fictional characters, but the paper does not specify their genre (e.g., anime, novel, film) or personality span (e.g., introverted vs. extroverted, heroic vs. villainous). If characters are concentrated in a single genre or share similar traits, the framework’s generalization to diverse role-playing scenarios (e.g., classical novel characters) remains unvalidated.
> ```
>
> We appreciate the reviewer’s insightful question. The table below shows the genres of works in which each character appears. Note that while there are some similar categories, CharacterRAG encompasses a broader range of genres. The detailed personality distribution for each character (e.g., Extroverted vs. Introverted) is shown in Figure 2.
>
> | Character |  Genre  |
> | :---: | :---: |
> | Anya Forger |  Action, Comedy, Drama, Slice of Life, Spy      |
> | Chika Fujiwara |  Gag, Romantic Comedy  |
> | Edward Elric        | Adventure, Fantasy    |
> | Frieren             | Fantasy, Adventure                             |
> | Hitori Gotoh        | Band, Comedy, Drama, Music, Slice of Life      |
> | Light Yagami        | Crime, Dark Fantasy, Noir, Thriller            |
> | Mao Mao             | Medical, Mystery, Palace Mystery                |
> | Megumin             | Comedy, Fantasy, Isekai                         |
> | Mikoto Misaka       | Cyberpunk, SF, Slice of Life                     |
> | Nina Iseri          | Band, Drama                                    |
> | Saitama             | Action, Comedy, Hero                     |
> | Son Goku            | Battle, Comedy, Fantasy           |
> | Tanjiro Kamado      | Action, Adventure, Historical Fantasy     |
> | Tobio Kageyama      | Sports, Volleyball                  |
> | Yui Hirasawa        | Band, School, Slice of Life               |
>
>
> ```
> The Attribute Extractor (AE) only extracts "Belief and Value" and "Psychological Traits," claiming they "directly influence behavior". But it does not explain why other attributes (e.g., "Social Relationships" or "Skill and Expertise," which also shape role responses) are excluded, nor provide comparative experiments to prove these two attributes are more critical for persona consistency. Is it specific to tasks like MBTI and BFI?
> ```
>
> Inspired by [1, 2], we assume that the attributes constituting a persona can be categorized into two groups: those that significantly influence the character's behavior, and those that primarily define the character's knowledge. The attributes that constitute a character’s knowledge can be sufficiently addressed using Retrieval-Augmented Generation (RAG), a general information retrieval approach. However, attributes that determine a character’s behaviors are difficult to resolve through retrieval, and are often absent from typical persona representations. For example, a question such as $\textit{“How did Anya Forger perform on her first midterm exam?”}$ can typically be addressed by conventional RAG methods. However, a question like $\textit{“Is Anya Forger more interested in complex and novel ideas than in simple and intuitive ones?”}$ is challenging for RAG to answer effectively.
>
> We evaluate whether the methods effectively utilize the attributes that constitute a character's knowledge through our manually constructed CharacterRAG, and we assess whether the methods make appropriate use of the attributes that influence a character’s behavior by employing psychological questionnaires such as MBTI and BFI, which are not explicitly included in the knowledge.
>
> ```
> The evaluation dataset is incomplete. The paper only evaluates tasks like MBTI and BFI, but role-playing involves many other dimensions. Is the method applicable to other role-playing tasks? For example, imitating a character's linguistic style or simulating a character's behavior.
> ```
>
> Thanks for raising this question. As the reviewer mentioned, there is a lack of diversity in evaluating whether RPA effectively utilizes the six attributes. However, there has been little research on RAG-based RPAs, and consequently, there is a lack of datasets available for building and evaluating such RPAs. Noting these aspects, we construct a high-quality dataset designed for building and evaluating RAG-based RPAs. We aim for this dataset to provide guidelines for future research, and we will make CharacterRAG dataset publicly available.
>
>
> ```
> How is L_max set?
> ```
>
> At each iteration, we load each paragraph that constitutes a persona, measure its length, and define the longest length among them as L_max.

---

> ### Author Response · Authors · 2025-11-25
>
> # [2/2]
> ```
> Why can finding the information of the k most similar chunks solve problems that are beyond the scope of the character's knowledge?
> ```
>
> As shown in Figure 1, previous RAG methods tend to excessively reuse irrelevant chunks when answering questions that go beyond the character’s knowledge. We address these issues through the use of ACTS, GS, and AE. The table below presents the ablation study results for each of the components.
>
> | Method |  $\sum \|d\|$ ($\downarrow$)  |  Accuracy ($\uparrow$)  |  Avg F1-Score ($\uparrow$)  |
> | :---: | :---: | :---: | :---: |
> | Naive RAG |  21  |  65.00\%  |  0.6146  |
> | Naive RAG + ACTS |  19  |  68.33\%  |  0.6524  |
> | Naive RAG + ACTS + GS |  15  |  75.00\%  |  0.7426  |
> | **Naive RAG + ACTS + GS + AE (Ours)** |  **9**  |  **85.00\%**  |  **0.8394**  |
>
> Results demonstrate that ACTS, GS, and AE are each highly proficient at addressing queries that exceed the character’s given knowledge. Notably, AE delivers the greatest performance enhancement, suggesting that extracting and leveraging character attributes as inputs is significantly more beneficial than merely supplying passages relevant to the user's question.
>
>
> We would like to express our sincere gratitude for the reviewer’s insightful comments. We hope that our responses have adequately addressed the concerns raised. If there are any additional questions or if further clarification is needed, we would be more than happy to provide further explanations.
>
>
> [1] Park et al., “CharacterGPT: A Persona Reconstruction Framework for Role-Playing Agents.” NAACL. 2025.
>
> [2] Chen et al., “Towards a Design Guideline for RPA Evaluation: A Survey of Large Language Model-Based Role-Playing Agents.“ ACL. 2025. Findings.

---

### Official Review · Reviewer_1rjH · 2025-11-01

**Soundness:** 3
**Presentation:** 3
**Contribution:** 3
**Rating:** 6
**Confidence:** 4

**Summary:**

The paper proposes a training-free RAG framework for role-playing agents that targets persona consistency even when user queries fall outside a character’s explicit knowledge. The apporoach has three stages: (1) Adaptive Context-aware Text Splitter creates optimally sized, overlapping chunks annotated with hierarchical headings; (2) Guided Selection iteratively filters retrieved chunks using an LLM to prefer passages from which traits can be inferred; and (3) Attribute Extractor derives beliefs/values and Psychological Traits from the selected chunks and feeds them as final context for response generation. The authors also release CharacterRAG, a 15-character, 976K-char persona corpus with 450 QA pairs. Experiments across close-source or open-source models report higher accuracy and lower hallucination scores than RAG baselines, and  human raters find attributions reasonable

**Strengths:**

The paper explicitly targets a common failure mode in RAG-based role-playing: when a user asks about aspects that are not explicitly in the persona, vanilla retrievers overuse low-relevance chunks and the agent hallucinates. The abstract and introduction motivate this crisply and position AMADEUS as training-free with three modules.

ACTS preserves hierarchical context with empirical support that maximizes summed similarity and minimizes variance; ACTS/ATS outperform standard splitters across embeddings.

CharacterRAG contains 15 fictional characters (976k characters) and 450 QA pairs with six attribute categories, constructed from the character’s viewpoint (editorial/meta information removed). This fills a gap for RAG-based RPAs. AMADEUS improves stronger performances.

**Weaknesses:**

CharacterRAG contains only 15 fictional characters, and much of the persona content is mined from Namuwiki; it remains unclear how well findings transfer to real people, evolving personas. Adding non-fictional or time-varying personas would strengthen claims.

While ACTS’s hierarchical extraction cost is noted (O(N)), the end-to-end latency and token/dollar costs (especially for GS/AE with large models) are not reported in detail across LLMs/datasets, limiting deployment guidance.

The related work should cover several fast-moving threads which is missing: (i) LongRAG’s long-unit/long-reader paradigm that challenges short-chunk assumptions; also add RankRAG, which unifies reranking with generation and is conceptually close to your Guided Selection. (ii) Role-playing/persona literature: evaluation benchmarks (CharacterEval; RoleLLM; InCharacter and a non-RAG), Non-RAG persona models (Persona-Adaptive Attention).

The w/o-RAG baseline shows non-trivial background knowledge, but the paper does not deeply analyze how much persona knowledge the base LLMs already encode or how AE/GS ablations isolate gains beyond retrieval.

**Questions:**

NA

---

> ### Author Response · Authors · 2025-11-25
>
> # [1/2]
>
> We sincerely appreciate reviewer 1rjH’s insightful comments and suggestions. Below, we address each of the points that have been raised.
>
> ```
> The related work should cover several fast-moving threads which is missing: (i) LongRAG’s long-unit/long-reader paradigm that challenges short-chunk assumptions; also add RankRAG, which unifies reranking with generation and is conceptually close to your Guided Selection.
> ```
>
> Although both methods are strong baselines, due to the lack of available implementations [3] or its reliance on long contexts resulting in high computational costs, which are beyond the scope of our study [2], we instead consider the following two baselines: RAPTOR [4] and Adaptive RAG [5]. To ensure a fair comparison and address the reviewer's comment, the component responsible for determining query complexity in Adaptive RAG is reimplemented to include both a module for assessing the relevance of queries and another for detecting hallucinations in responses. Meanwhile, RAPTOR recursively builds a tree with varying levels of summarization by iteratively applying embedding, clustering (using Gaussian Mixture Model), and summarization steps. It should be noted that we use 60 MBTI questions to examine whether RAG-based RPA can appropriately answer questions that fall outside the scope of its existing knowledge.
>
> |  |  **Naive RAG**  |  **CRAG**  |  **LightRAG**  |  **Adaptive RAG**  |  **RAPTOR**  |  **AMADEUS (Ours)**  |
> | :---: | :---: | :---: | :---: | :---: | :---: | :---: |
> | $\sum \|d\|$ ($\downarrow$) |  21  | 19 | 21 | 15 | 14 | **9** |
> | Accuracy ($\uparrow$) |  65.00\%  | 68.33\% | 65.00\% | 75.00\% | 76.67\% | **85.00\%** |
> | Avg F1-Score ($\uparrow$) |  0.6146  | 0.6448 | 0.5344 | 0.6832 | 0.6925 | **0.8394** |
>
>
> Results demonstrate that, in response to user queries, not only utilizing appropriate knowledge but also approximating character attributes, such as Psychological Traits, based on the given information is the most effective approach. We believe that these findings have the potential to open up new avenues in the field of RAG-based RPA, which remains relatively underexplored.
>
> ```
> The related work should cover several fast-moving threads which is missing: (ii) Role-playing/persona literature: evaluation benchmarks (CharacterEval; RoleLLM; InCharacter and a non-RAG), Non-RAG persona models (Persona-Adaptive Attention).
> ```
>
> Since CharacterEval, RoleLLM, and InCharacter are highly valuable as evaluation benchmarks, they are not well-suited for building and quantitatively evaluating RAG-based role-playing agents. We note that there has been little research on role-playing based on RAG, and therefore we built CharacterRAG with high quality and plan to release it to the public. We believe that the results of these two approaches will be greatly beneficial for future works on hybrid methods.
>
>
> ```
> The w/o-RAG baseline shows non-trivial background knowledge, but the paper does not deeply analyze how much persona knowledge the base LLMs already encode or how AE/GS ablations isolate gains beyond retrieval.
> ```
>
> | Method |  $\sum \|d\|$ ($\downarrow$)  |  Accuracy ($\uparrow$)  |  Avg F1-Score ($\uparrow$)  |
> | :---: | :---: | :---: | :---: |
> | Naive RAG |  21  |  65.00\%  |  0.6146  |
> | Naive RAG + ACTS |  19  |  68.33\%  |  0.6524  |
> | Naive RAG + ACTS + GS |  15  |  75.00\%  |  0.7426  |
> | **Naive RAG + ACTS + GS + AE (Ours)** |  **9**  |  **85.00\%**  |  **0.8394**  |
>
> The table above demonstrates the results of ablation studies using GPT-4.1 ("gpt-4.1- 2025-04-14"), evaluated on 60 MBTI questions. We find that ACTS, GS, and AE are all highly effective at handling questions that go beyond a character's knowledge. Note that AE achieves the largest performance improvement, highlighting that extracting and incorporating character attributes as input is far more effective than simply providing chunks related to the user query.

---

> ### Author Response · Authors · 2025-11-25
>
> # [2/2]
> ```
> While ACTS’s hierarchical extraction cost is noted (O(N)),
> the end-to-end latency and token/dollar costs (especially for GS/AE with large models) are not reported in detail across LLMs/datasets, limiting deployment guidance.
> ```
>
> |  |  **Naive RAG**  |  **CRAG**  |  **Adaptive RAG**  |  **RAPTOR**  |  **AMADEUS (Ours)**  |
> | :---: | :---: | :---: | :---: | :---: | :---: |
> | Total Latency (s) |  4048.61  | 9,214.99 | 36929.92 | 17155.33 | 17173.02 |
> | Total Tokens |  486,058  | - | 12,187,670 | 2,485,627 | 5,649,665 |
> | Total Cost ($) |  1.87  | 6.67 | 28.26 | 9.88 | 14.05 |
>
> We evaluate each method by measuring the total latency, the number of tokens consumed, and the overall cost across all 60 MBTI questions. To align with the experimental results in the paper, we employ GPT-4.1 (“gpt-4.1- 2025-04-14“) as the backbone. RAPTOR additionally includes the cost incurred for tree construction.
>
> Results show that although GS effectively utilizes the character’s persona there is still room for further improvement. The probability that GS fails in all search iterations trials (30) is 15.33%. In terms of GS processing speed, Gemma3-27B requires approximately 0.1 seconds for a single request in the vLLM environment, whereas GPT-4.1 takes around 0.49 seconds. Meanwhile, the total latency of LightRAG was 25,065 seconds, while that of GraphRAG was approximately 81,900 seconds.
>
> ```
> CharacterRAG contains only 15 fictional characters, and much of the persona content is mined from Namuwiki;
> it remains unclear how well findings transfer to real people, evolving personas. Adding non-fictional or time-varying personas would strengthen claims.
> ```
>
> Thanks for raising this question. Indeed, in RAG-based RPA, it is possible to express a character whose persona evolves over time by modifying the persona according to the character’s actions within the story [1]. In future work, we will explore how such methods can be applied to real people and investigate ways to enable the evolution of personas over time.
>
> We sincerely appreciate the reviewer’s thoughtful comments. We hope that our responses have sufficiently addressed the concerns raised. Should the reviewer have any further questions or require additional clarification, we would be glad to provide further discussion.
>
> [1] Park et al., “CharacterGPT: A Persona Reconstruction Framework for Role-Playing Agents.” NAACL. 2025.
>
> [2] Zhao et al., “LongRAG: A Dual-Perspective Retrieval-Augmented Generation Paradigm for Long-Context Question Answering.“ EMNLP. 2024.
>
> [3] Yu et al., “RankRAG: Unifying Context Ranking with Retrieval-Augmented Generation in LLMs.“ NeurIPS. 2024.
>
> [4] Jeong et al., “Adaptive-RAG: Learning to Adapt Retrieval-Augmented Large Language Models through Question Complexity.“ NAACL. 2024.
>
> [5] Sarthi et al., “RAPTOR: Recursive Abstractive Processing for Tree-Organized Retrieval.“ ICLR. 2024

---

### Author Response · Authors · 2025-12-03
**General Answer to All Reviewers, AC, SAC, and PC**

Dear Reviewers, AC, SAC, and PC,

We sincerely appreciate the thoughtful comments and constructive questions provided by the reviewers. We are grateful for the time and effort you invested in evaluating our manuscript, and we deeply value your recognition of the significant work encompassed in this work.

We have carefully revised our manuscript in response to all the comments and suggestions provided by the reviewers, as well as incorporating further improvements made since the initial submission. We believe that these modifications have significantly enhanced the quality and clarity of the paper. For your convenience, we have summarized the major changes below, which we believe address the primary concerns raised by the reviewers:

- **Ablation Analysis** (for Reviewer 1rjH, DAgg, EE4o, gUX3). We report the impact of each stage of our framework on performance when 15 characters are confronted with out-of-knowledge questions.

- **Data Quality Inspection** (for Reviewer 1rjH, DAgg, EE4o): Noting that there is little research on RAG-based RPAs, we emphasize the fact that we construct the CharacterRAG dataset, which is necessary for building and evaluating RPAs. In addition, we report that the characters appear in a wide range of genres, and we clarify the dataset construction process in more detail.

- **Mechanism Verification** (for Reviewer 1rjH, gUX3): We demonstrate through additional experiments that extracting characters' attributes from the knowledge chunks and using them as context is significantly more effective than simply using the knowledge chunks themselves.

- **Qualitative Analysis** (for Reviewer DAgg, EE4o, gUX3): We improve interpretability by clarifying the results of each stage of our framework with illustrative examples.

- **Experimental Clarifications** (for Reviewer 1rjH, EE4o, gUX3):
    - **Efficiency**: We report the total latency and cost of our method and the baselines to provide deployment guidance.
    - **Prompt**: We clarify the prompts used in our framework.
    - **Evaluation Protocols**: By referencing prior research in our selection of psychological questionnaires (e.g., MBTI and BFI) for the experiments and in the establishment of ground truth, we strengthen the validity and credibility of our manuscript.

---
We would like to express our heartfelt appreciation to all the reviewers, AC, SAC, and PC for dedicating their valuable time to evaluate our work.

Furthermore, we are deeply grateful to the reviewers for their insightful comments and constructive feedback, which have been instrumental in enhancing the quality of this paper.


Best regards,

The authors

---

### Meta-Review · Area_Chair_rjGt · 2025-12-19

**Summary:**

The reviewers are concerned that the paper has significant weaknesses in its contribution, clarity, and empirical validity. The work reads more like a dataset-specific engineering report than a research paper with a well-motivated problem and clear methodological innovation. In particular, framing personality modeling, which is a largely global and stable attribute, within a RAG framework is not convincingly justified, as simpler full-context LLM summarization may be more appropriate. The paper also suffers from poor writing and organization, with many essential details missing. The construction and validation of the CharacterRAG dataset, the concrete implementation of ACTS, Guided Selection, and the Attribute Extractor, as well as the interpretation of key figures, are all insufficiently explained, making the approach difficult to understand and reproduce. Empirically, the evaluation is overly narrow and relies heavily on MBTI and BFI scores, whose ground-truth assignment is unclear and potentially unreliable, with no evidence that the method benefits broader downstream tasks or avoids hallucinated traits. Moreover, the approach appears highly tailored to the proposed dataset and depends on computationally expensive LLM-based reranking, raising concerns about fairness, scalability, and generalizability. Based on these concerns, I recommend rejection.

**Reviewer Concerns:**

Although the authors attempted to address some reviewer concerns in the rebuttal by providing additional explanations of the method details and adding supplementary experimental results, these responses mainly clarified implementation choices rather than resolving the core issues. In particular, the rebuttal helped partially address questions about how the main modules are implemented and how certain experimental settings are configured. However, several major concerns remain outstanding, including the fundamental problem framing of using RAG for personality modeling, the limited and MBTI-centric evaluation scope, the unclear validity of personality ground truths, and the dataset-specific nature and scalability of the proposed approach. As a result, while the rebuttal improved clarity on some technical points, it did not substantially change the overall assessment of the paper’s quality or contribution.

**Reviewer Scores:**

I do not believe the reviewers would have changed their scores. Their concerns are fundamental and consistent across reviews, focusing on the paper’s problem framing, lack of methodological clarity, dataset-specific design, and limited empirical gains. These issues are unlikely to be resolved through discussion alone, as they reflect structural weaknesses rather than misunderstandings or minor points of clarification.

---

### Decision · Program_Chairs · 2026-01-26

Reject